# SARS-CoV-2 Gut-Targeted Epitopes: Sequence Similarity and Cross-Reactivity Join Together for Molecular Mimicry

**DOI:** 10.3390/biomedicines11071937

**Published:** 2023-07-07

**Authors:** Aaron Lerner, Carina Benzvi, Aristo Vojdani

**Affiliations:** 1Chaim Sheba Medical Center, The Zabludowicz Research Center for Autoimmune Diseases, Ramat Gan 52621, Israel; carina.ben.zvi@gmail.com; 2Research Department, Ariel University, Ariel 40700, Israel; 3Immunosciences Lab., Inc., Los Angeles, CA 90035, USA; drari@msn.com

**Keywords:** SARS-CoV-2, COVID-19, gut, intestine, gastrointestinal tract, sequence similarity, cross-reactivity, molecular mimicry, exposome

## Abstract

The gastrointestinal tract can be heavily infected by SARS-CoV-2. Being an auto-immunogenic virus, SARS-CoV-2 represents an environmental factor that might play a role in gut-associated autoimmune diseases. However, molecular mimicry between the virus and the intestinal epitopes is under-investigated. The present study aims to elucidate sequence similarity between viral antigens and human enteric sequences, based on known cross-reactivity. SARS-CoV-2 epitopes that cross-react with human gut antigens were explored, and sequence alignment was performed against self-antigens implicated in enteric autoimmune conditions. Experimental SARS-CoV-2 epitopes were aggregated from the Immune Epitope Database (IEDB), while enteric antigens were obtained from the UniProt Knowledgebase. A Pairwise Local Alignment tool, EMBOSS Matcher, was employed for the similarity search. Sequence similarity and targeted cross-reactivity were depicted between 10 pairs of immunoreactive epitopes. Similar pairs were found in four viral proteins and seven enteric antigens related to ulcerative colitis, primary biliary cholangitis, celiac disease, and autoimmune hepatitis. Antibodies made against the viral proteins that were cross-reactive with human gut antigens are involved in several essential cellular functions. The relationship and contribution of those intestinal cross-reactive epitopes to SARS-CoV-2 or its potential contribution to gut auto-immuno-genesis are discussed.

## 1. Introduction

Coronavirus disease 2019 (COVID-19) has been a global pandemic health concern for the last three years. Its high infectivity, vast geographical distribution, wide clinical heterogenicity, prognosis, mortality, and short/long-term outcome drove the clinical and scientific communities to study the disease’s viral geoepidemiology, mode of action, and potential therapeutic modalities in order to overcome the pandemic. One recently reported aspect of COVID-19 is the plethora of autoimmune diseases (ADs) and corresponding autoantibodies that are continuously being reported to be associated with the SARS-CoV-2 virus [1,2,3,4,5]. SARS-CoV-2 is now being described as an auto-immunogenic virus [6,7,8,9,10,11,12]. Ads have been recognized as part of the long COVID syndrome, and many are concerned regarding a future surge in the incidence of those conditions [13].

It is well established that genetic background is crucial for the development of Ads, but environmental factors are pivotal in their clinical expression [14,15,16,17]. In this sense, diet and hygiene [18], gut microbial antigenicity [19], multiple infectious agents [17,20,21], various vaccines [22,23], food processed additives [24,25,26], toxic agents [16], and even chemicals and drugs [27] have been suggested to take part in the environmentally induced autoimmunity of ASIA (autoimmune/inflammatory syndrome induced by adjuvants) [14,28]. Focusing on the gastrointestinal tract (GIT), many of the above-mentioned environmental factors dwell in the lumen and are associated with ADs [16,17,18,19,20,21,25,26,29]. Intriguingly, SARS-CoV-2 infects the human intestine [30,31,32,33,34,35] and recently was enumerated as a “new runner in the gastrointestinal tract” [30].

Many mechanisms for driving gut-originated local and systemic autoimmunity have been suggested to operate in the human GIT. Increased gut permeability can result in leaky gut syndrome [29], posttranslational modification of naïve proteins [36], dysbiosis and its harmful mobilome [37], and horizontal gene transfer [38]. All those enteric events irradiate peripherally and might induce systemic autoimmunity. Several mechanisms drive autoimmunity [1]: molecular mimicry [19,39,40,41], epitope spreading [42,43], bystander activation [44,45] violation of biological barriers in immune-privileged organs [46], polyclonal activation of lymphocytes as a result of the action of superantigens [47], and finally, immune cross-reactivity between non-self- and self-antigens [48,49,50,51,52]. Several of those modes of action were even suggested to drive SARS-CoV-2-associated autoimmunity [1,11,12,40,47,48,49,50,53]. A major under-studied aspect of the SARS-CoV-2-induced ADs is the molecular mimicry between the virus and the enteric epitopes. The present study aims to elucidate sequence similarity between shared virus antigens and the human enteric peptides’ sequences and compare it to known cross-reactivity between the two.

## 2. Materials and Methods

To perform sequence alignment between self-enteric antigens and SARS-CoV-2 epitopes, assays of enteric antigens and SARS-CoV-2 epitope databases were explored. IEDB was searched with the following keywords: “Linear Epitope”, “SARS-CoV-2 Organism”, and “Human Host”. Assays included: “T cells,” “B cells”, “HLA-I”, and “HLA-II” and were rated as “Positive Assays”. In this search, 10,794 SARS-CoV-2 epitopes were retrieved. IEDB was also used to identify antigens that are implicated in enteric ADs. Out of 12,957 self-antigens, 32 were detected in IEDB, and their derived sequence was extracted from the UniProt Knowledgebase [54]. A Pairwise Local Alignment tool, EMBOSS Matcher [55], was employed to explore local similarities between the aggregated SARS-CoV-2 epitopes and those auto-antigens. This tool implements an algorithm that is based on Bill Pearson’s Lalign application, version 2.0u4 (February 1996). Using a Python script that can be found at https://www.ebi.ac.uk/Tools/emboss/ (accessed on 30 December 2022), EMBOSS Matcher was applied to each of the SARS-CoV-2 epitopes against the sequence of the 32 enteric antigens. According to publications by Kanduc D. [56,57,58], a peptide motif of five amino acids (AAs) can act as a minimal antigenic determinant in humoral and cellular immune recognition, particularly when these five-residue peptides are in the central core of T-cell epitopes of 8 to 20 AA long. A recent example of how sequence similarity between foreign and self-proteins can induce molecular mimicry was found between Epstein–Barr virus nuclear antigen 1 (EBNA1) and the glial cell adhesion molecule, a protein of the central nervous system [59], in which five out of six AAs were identical. Therefore, in our research, the aligned peptides’ cut-off was kept at a minimum of six identical AAs and at peptide length ≥ seven AAs. The result was 58 similar sequences. All SARS-CoV-2 epitopes were captured in IEDB from experimental assays that were published in scientific literature. However, since EMBOSS Matcher randomly extracted similar peptides out of the enteric whole proteins, an additional analysis was required to assess their immunological potential reactivity. Although computational methods continue to improve, they cannot replace experimental validation conducted through laboratory assays [60]; therefore, an additional search was required in IEDB for epitopes that (1) harbor one of those enteric similar sequences, (2) were experimentally validated as auto-immunogenic in various assays, and (3) are derived from antigens that are implicated with at least one of the seven enteric ADs and preferably antigens that exhibited cross-reactivity with SARS-CoV-2 proteins. To eliminate redundancy, overlapping epitopes were removed so that each enteric protein has one epitope per similar sequence. The abovementioned methodology is presented as a flowchart in Figure 1.

## 3. Results

Cross-reactivity between human intestinal and hepatic antigens and SARS-CoV-2 peptides was studied by analyzing the publications of Vojdani A. et al. on the topic [16,17,48,49,50]. Multiple human self-proteins cross-react with the virus epitopes. The strongest to moderate reaction between the spike, envelope, membrane, and nucleoprotein of the virus was detected against intestinal epithelial cells, mitochondrial antigen (M2), the liver microsome, and tissue transglutaminase. A weaker but still positive cross-reaction was shown for intestinal barrier proteins such as actin, occludin, zonulin, β-catenin, α-myosin, and S100B [49]. A comparatively weak reaction was detected against human collagen and histone, two integral proteins of the human gut and liver.

In searching for autoantigenic enteric epitopes in experimental studies, seven ADs were examined: autoimmune gastritis, autoimmune hepatitis (AIH), autoimmune pancreatitis, celiac disease (CD), primary biliary cholangitis (PBC), and two categories of inflammatory bowel disease (IBD), ulcerative colitis (UC), and Crohn’s disease. Out of the 58 pairs of similar sequences, 10 were also identified as cross-reactive against four specific SARS-CoV-2 proteins, as reported by Vojdani A. et al. [16,17,48,49,50]. Interestingly, there are additional seven similarities between those intestinal antigens and SARS-CoV-2 ORF1ab polyprotein. Notably, this protein is cleaved to form nonstructural proteins [61]. Since it was not explored for cross-reactivity, it is not included in Table 1, Table 2 and Table 3; instead, this protein is included in the Appendix A.

Table 1 presents the 10 sequence similarities driven by seven antigens. One antigen relates to UC; three relate to PBC; one relates to CD; and five relate to AIH, out of which one is implicated with two ADs. The similarity of paired sequences is displayed in red, (human on top of SARS-CoV-2), and the isolated AA mismatches are marked in black. The alignment cut-off was kept at a minimum of six identical AAs, and peptide length > seven AAs. The resulting enteric sequences are presented in Table 2 highlighting the antigens’ functionality and their implications in various ADs. Table 3 presents the epitopes that had immunoreactive validation in experimental assays, as curated in IEDB, and were found to harbor one of the paired sequences similarities. Each epitope sequence is listed with its reference to the related immune-biological assay used. In addition to the 10 pairs of sequences described in Table 1, 48 pairs whose epitopes did not match with cross-reactive antigens are included in the Appendix A.

## 4. Discussion

SARS-CoV-2 causes COVID-19 disease which geo-epidemiologically represents a continuing pandemic, impacting most countries. The contagious pandemic might cause a severe disease, resulting in extensive morbidity and high mortality, mainly affecting the elderly and various vulnerable populations. Interestingly, the name of the offending virus (Severe Acute Respiratory Syndrome Coronavirus 2) implies that the target organ is the upper respiratory ways and lungs. However, the GIT and its affiliated organs are no less end-target organs. In previous recent publications, it was observed that the virus angiotensin-converting enzyme 2 (ACE2) and TMPRSS2 receptors have a wide enteric distribution, the gut is heavily infected, there is stool SARS-CoV-2 shading, and the affected patients are often symptomatic [30,31,32,33,34,35]. As evidence for the substantial GIT contamination, fecal–oral transmission is suggested [105,106,107,108].

One of the major pathophysiological pathways by which the virus operates in COVID-19 involves hyper-stimulation of the immune system and the resulting cytokine storm [109]. Being aware of the potential consequences, the autoimmune immunologists suspected a flare of ADs in the post-COVID period. Their suspicions were confirmed, and SARS-CoV-2 is described, currently, as an auto-immunogenic virus [2,3,4,5,6,7,8,9,10,11,12,53,110]. Several pathways were suggested to operate in the SARS-CoV-2-AD cross-talks [1,11,12]. However, the most accepted and reported mechanism is molecular mimicry between the viral epitopes and human antigens [9,40,41,53,111]. Molecular mimicry is defined as the process where sequence similarities between foreign and self-peptides are sufficient to result in the cross-activation of B or autoreactive T cells, induced by pathogen-derived peptides [39,112]. The definition can be expanded to other environmental factors, including food components, toxins, or many other exposomes, where cross-reactive antibodies were identified [16,17,48,49,50,51,52]. The contribution of the present work fills the gap of the under-explored cross-talks between cross-reactivity and sequence similarity between the SARS-CoV-2 virus and the human GIT and its associated organs. Both phenomena join together to drive molecular mimicry (Figure 2 and Figure 3). Cross-reactive antibodies and sequence similarity between viral and self-antigens display a potentially major role in the pathophysiology of ADs [16,17,39,112]. Several studies described those phenomena in mammalians or on the whole human body level; however, the present report concentrates, for the first time, exclusively on shared SARS-CoV-2 and GIT epitopes related to specific ADs and not on shared epitopes with various anti-COVID 19 vaccines.

After obtaining the enteric epitopes, similar sequence pairing was detected in 58 different SARS-CoV-2 epitopes, displaying 100% core identity. Intriguingly, those shared sequences are related to five gut-associated ADs (Table 1 and Appendix A). Most of the shared sequences, 31 out of 58, are related to AIH; 10 are related to IBD (UC, and Crohn’s Disease); 13 are related to PBC; and eight are related to CD (some are implicated with two ADs). It appears that the human hepatic ADs (AIH, PBC) shared more epitopes, compared to the human-affected bowel ones (CD, IBD), or mythologically, the IEDB platform contains more hepatic epitopes a priory. In reality, the liver is a target organ in COVID-19 disease [113,114,115]. AIH was described post-SARS-CoV-2 disease even after vaccination [116,117,118], and PBC is also related to COVID-19 [119]. Analyzing the potential mechanistic aspects of those human proteins, it is interesting to discover that all of them had potential roles and might be involved in the induction of each of those five ADs (Table 2 and Appendix A). The potential pathological involvement of those proteins, being mutated or experiencing loss of function under certain environmental pressures, such as COVID-19, is far from being elucidated. In general, they are involved in many cellular, sub-cellular, and biological functions in the human intestinal tract. The list includes cell division, enteric tight junction functional integrity, nuclear matrix, and filament cytoskeleton morphogenesis, muscle contractility, immune modulation and autoantibody production, and epigenetic modification, all of which are involved in CD and AIH (Table 2 and Appendix A). The immunogenicity of similar sequences was assessed by searching IEDB for epitopes that harbor these sequences and which were experimentally validated. These 58 corresponding epitopes, their matching sequences, and their various types of assays are detailed in Table 3 and Appendix A. Notably, some epitopes were detected with more than one assay, which implies their prevalence. Since viral versus self-antigens have been studied for different aspects, most of the human assays were performed on specific HLA-I/II alleles, mainly for auto-immunogenic evaluation, while SARS-CoV-2 peptides are more diverse and contain many T-cells/B-cells epitopes. This can also be explained due to the fact that all the viral assays were performed within the last three years, with readily available advanced equipment.

Cross-reactivity between the SARS-CoV-2 spike, envelope, and nucleoprotein and the intestinal and hepatic antigens was extensively reported in a study by Vojdani A. et al. [48]. The accepted assumption is that following infection or COVID-19 vaccination, an immune response is triggered against the viral antigens that cross-react with human gut self-antigens which share sequence similarity with viral constituents, resulting in an auto-immunogenic cascade, eventually progressing to an AD phenotype. Interestingly, several intestinal barrier proteins were described to cross-react with the spike, nucleo, envelope, and membrane protein of the SARS-CoV-2 virus, and enhanced intestinal permeability is a major pathway that drives ADs [25,29], including IBD, CD, AIH, and PBC. Of note, several major tight junction proteins (actin, occludin, zonulin, β-catenin, α-myosin, and S100B) shared cross-reactive antibodies with SARS-CoV-2. While all of them are integral structural components of the intestinal tight junction, the last one merits some explanation. Intestinal glial cells were shown to secrete S100B [120]. Local S100B over-expression is associated with enhanced inflammation in the human gut, thus playing a major role in gut–brain-induced inflammation and autoimmunity [121]. More so, those cross-reactive antibodies might lead to other leaky barriers in organs such as the lung or brain in genetically susceptible individuals [122]. Intriguingly, COVID-19-induced intestinal immune events might irradiate peripherally to induce ADs related to remote organs or play a role in the induction of other chronic diseases [123], many of which represent significant risk factors for enhanced morbidity and mortality in the current pandemic [124]. Those cross-reactive antibodies, by disrupting enteric permeability, might be responsible for multiple extra-intestinal manifestations of COVID-19, including hepatic autoimmune diseases, in the gut–liver axis frame [125]. Consequently, leaky gut induced by cross-reactive antibodies might enhance viral spreading to other organs and potentiate the cytokine storm [126].

The cross-reactivity with human hepatic protein, namely, liver microsomal and mitochondrial (M2) proteins, two pivotal autoantigens in AIH and PBC, respectively [71,72,127], might have paramount importance. Tissue transglutaminase (tTG) is the autoantigen that drives CD [128,129]. Indeed, all the last three proteins, the liver microsomal, the M2, and the transglutaminase-2 (tTG2) proteins, share cross-reactivity and sequence similarity, thus reinforcing their molecular mimicry with the SARS-CoV-2 structural components. Furthermore, the parallel detection, for the first time, of cross-reactive antibodies and sequence identity between the COVID-19 virus and the intestinal tight junction proteins, namely, actin, indicates again the connection to peptide mimicry autoimmunity progression.

SARS-CoV-2 spike protein and nucleoprotein reacted with tTG2. Since this enzyme has a wide distribution with almost all cell types in the body expressing it, the cross-reactive antibodies can potentially react with any cell that expresses the enzyme. This means that any remote organ, not just the intestines, might be affected. Even the brain can be affected, as was recently suggested for neurodegenerative conditions [130]. Another gastrointestinal protein, namely, intestinal epithelial cell antigen, reacted against SARS-CoV-2 envelope protein, using rabbit polyclonal antibody [49], thus augmenting an additional enteric target that might be affected by the SARS-CoV-2 virus cross-reactive antibodies.

To further pursue proof for this concept, the present study aimed to determine whether human monoclonal antibody, which mimics natural antibodies produced by the immune system to fight the SARS-CoV-2 virus, will react with several enteric and hepatic tissue antigens that are involved in the corresponding gut–liver affected ADs. Combining the observations of Vojdani et al. on cross-reactivity [16,17,48,49,50,51,52] with the presently reported sequence identity, several conclusions, representing the study contributions, can be drawn:SARS-CoV-2 spike, nucleo, envelope, and membrane protein cross-react with major enteric and hepatic self-antigens.Sequence similarity exists between multiple coronavirus and intestinal/hepatic epitopes.Cross-reactive autoantibodies and sequence similarity are major potential drivers of molecular mimicry in the auto-immunogenic avalanche.The combined shared cross-reactive and sequence identical core in gut-associated epitopes further strengthens the connection to molecular mimicry [39,131]. The two mechanisms detected major essential proteins that represent specific autoantigens for CD, AIH, and PBC or contribute to leaky gut by disrupting tight junction functionality, thus becoming directly involved in the corresponding auto-immuno-genesis [29].

Cross-reactive antibodies against the mitochondrial M2 antigen might have pathophysiological and clinical consequences. Mitochondrial M2 had strong to very strong reactions with all four SARS-CoV-2 protein antibodies [49]. Pathophysiological mitochondrial dysfunction is part of the multi-organ dysfunction or failure associated with COVID-19 disease and post-COVID syndrome [132,133]. M2 is part of the pyruvate dehydrogenase complex, and 90–95% of the PBC-affected patients are positive for the anti-mitochondrial antibodies [134]. Taking into account the present dual positivity between the M2 cross-reactive antibodies [49] and the presently shared sequence similarity between M2 and the SARS-CoV-2 epitopes, one wonders what role is played by the mitochondrial M2 dysfunction in AIH [135] and PBC [119,135] induction or exacerbation.

The plethora of those matches between SARS-CoV-2 sequences and human tissues might explain why monoclonal antibodies made against SARS-CoV-2 proteins reacted with so many tissue antigens. It should be noted though, that the present study was limited to the intestine and its associated organs, namely, the liver and the pancreas. Moreover, since the identification of general cross-reactive antibody responses was targeted at the level of antigens, while sequence similarity was inspected on specific epitopes, there is no guarantee that those antibodies cross-react within the same epitopes. Furthermore, the issue of linear/conformational epitopes was not part of the experimental design of the present or other studies [48,49,50,110]. In vivo, the conformational epitopes are much more relevant for the production of monoclonal neutralizing antibodies and targeted for autoantibody secretion in ADs [136]. The results may indicate that the SARS-CoV-2 antibodies reacted against conformational epitopes in the tissue antigens. Our study design did not specifically include analyses that would capture conformational or non-linear epitopes, but any of the tissue sequences that were found to match with the viral sequences, especially the highly recurring ones, could possibly be conformational epitopes. Conformational epitopes are not only important in the production of monoclonal neutralizing antibodies, but they could also be major targets of autoantibody production in ADs [137,138]. Other antigens among our list that had moderate or greater reactions with SARS-CoV-2 may also have sequences in their structure that could potentially be triggers of autoimmunity and likewise deserve additional attention and study. Further investigation to identify the specific cross-reactive epitopes will require specific peptide fragment inhibition studies as well as computational and immunogenicity predicational modeling including HLA-I/II alleles affinity. More precise identification of conformational autoepitopes is needed to clarify the role of SARS-CoV-2 in autoimmunity. Furthermore, very similar to earlier studies by Vojdani et al. [48,49], antibodies made against various SARS-CoV-2 proteins should be applied to the peptides that were shown to mimic various gut-associated epitopes described in this current study. The binding of these SARS-CoV-2 antibodies to these gut-associated epitopes would validate the findings of this study experimentally. Practically, this would require the synthesis of these gut-associated epitopes and their reaction with the SARS-CoV-2 antibodies, which is not within the scope of this current database study.

The present study might evoke concerns regarding anti-COVID-19 vaccinations. Potentially, the vaccines can induce cross-reactive antibodies against human tissue antigens, and the shared sequences may potentiate molecular mimicry between the vaccine-induced immunogenic epitopes and human ones, thus increasing the incidence of ADs [22,139,140]. However, vaccine-related cross-reactivity may not always necessarily be a bad thing. In 2020, Reche et al. decided to explore the question of why children are apparently largely spared from the ongoing COVID-19 epidemic [141]. The study found that combined vaccines for treating diphtheria, tetanus, and pertussis infectious diseases (DTaP vaccine) were significant sources of potential cross-reactive immunity to SARS-CoV-2. Reche et al. concluded that the DT antigens in combination DTaP vaccines are likely to keep children safe from COVID-19 worldwide. In the search for peptide matches with common pathogens, Reche et al. also found that, in addition, Bacillus Calmette-Guerin (BCG), had a staggering 3807 epitopes that matched with SARS-CoV-2. BCG is used against tuberculosis, and the Reche team noted that countries that implement BCG vaccination have fewer COVID-19 cases [141]. Thus, with regards to DTaP and BCG, cross-reactivity may mean cross-protective immunity. In support of Reche’s findings, very recently Vojdani et al. reacted monoclonal and polyclonal antibodies made against SARS-CoV-2 spike protein and nucleoprotein with different bacterial and viral antigens as well as the DTaP vaccine [142]. They found that SARS-CoV-2 spike protein and nucleoprotein reacted most significantly with the DTaP vaccine. The findings indicated that the cross-reactivity elicited by DTaP vaccines may indeed be keeping some individuals safe from COVID-19.

Finally, the present study documented, for the first time, cross-reactivity and sequence similarity between the SARS-CoV-2 virus and the gut-associated immunogenic epitopes. Although epitope sharing between SARS-CoV-2 and gut-targeted proteins is a very important factor, by itself it is not enough for the production of cross-reactive antibodies against autoantigens. This is why in addition to molecular mimicry, Vojdani emphasized the importance of the reaction of polyclonal and monoclonal antibodies made against SARS-CoV-2 proteins with a variety of human tissue antigens [17,48,49,52,142]. The present report further substantiates and confirms, thus, contributing to the importance of sequence similarity and cross-reactivity in auto-immuno-genesis. Those two phenomena are crucial for molecular mimicry, a major mechanism for the induction of ADs [23].

The topic is in its infancy, and only the tip of the iceberg is visible. Currently, we do not know how the emergence of SARS-CoV-2 variants and their effect on protein structure may influence this similarity and cross-reactivity with various tissue antigens, including those associated with the gut. It is hoped that our result will stimulate more studies in order to clarify this enigma and further resolve the mosaic of autoimmunity [143,144]. It still remains to elucidate how exactly viruses could shape autoimmunity.

## 5. Conclusions

Several articles have suggested that molecular mimicry between SARS-CoV-2 and human proteins drives autoimmunity. They postulate a connection between this mimicry and the multi-organ distribution of COVID-19, many of them beyond the respiratory system. In the present study, the known cross-reactive antibodies combined with the sequence core similarity between the virus and the GIT further strengthen the autoimmune pathway of molecular mimicry. It should be stressed that the present exploration deals with the SARS-CoV-2 and the human GIT autoimmune cross-talks and not with the potential, mRNA-based, anti-virus vaccination immune consequences. Since post-vaccination ADs are increasingly being reported, public health safety should be a prime concern to vaccinate against the virus, hence minimizing its harmful effects, including avoiding a surge in the incidence of ADs. Our findings of cross-reactivity and sequence similarity between the SARS-CoV-2 viral proteins and human tissue antigens provide support for the role of mimicry in autoimmunity. More extensive research on this subject can greatly aid in the battle against not just pandemics such as COVID-19 but also their various gut-associated autoimmune diseases, some of which could last a lifetime.

## Figures and Tables

**Figure 1 biomedicines-11-01937-f001:**
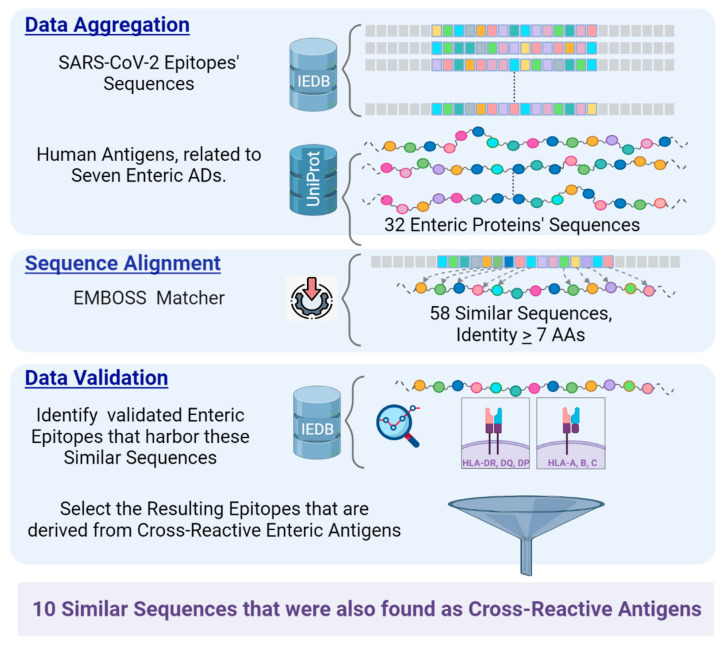
A graphical representation of the workflow. **Data Aggregation**: SARS-CoV-2 epitopes were extracted from IEDB, and human antigens that are implicated in enteric ADs were depicted. UniProt was searched to retrieve proteins sequences of the enteric self-antigens. **Sequence Alignment**: Emboss Matcher was employed; 58 Similar Sequences were found with a cut-off of at least 6 identical AAs and peptide length > 7. **Data Validation**: IEDB was searched to validate that the assayed enteric epitopes harbor those peptide sequences. Out of those, 10 were part of antigens that were previously identified to cross-react with SARS-CoV-2 antigens.

**Figure 2 biomedicines-11-01937-f002:**
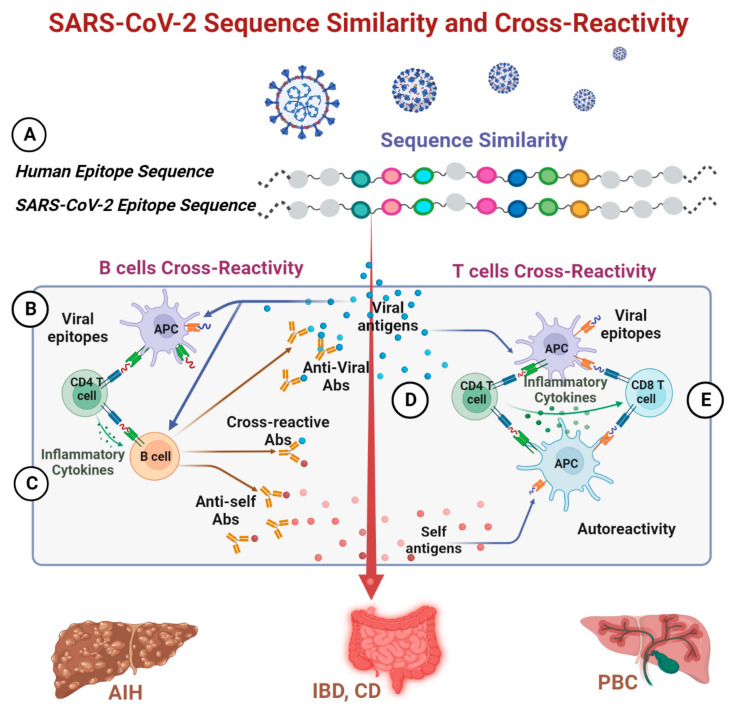
Schematic presentation of sequence similarity that leads to cross-reactivity between SARS-CoV-2 and gut-associated ADs. (**A**) Sequence similarity between a SARS-CoV-2 epitope and a Human epitope. (**B**) Interaction of viral antigens with immune cells and activating an adaptive immune system. (**C**) Cross-reactivity at B cell level when clonal antibodies bind to viral epitopes and to similar self-epitopes. (**D**) Cross-reactivity at the T cell level involves the recognition of viral epitopes and similar self-epitopes by the same CD4 T cell. CD4 T cells initiate an immune response against SARS-CoV-2 when APC present viral epitopes on HLA-II, but the same T cells have autoreactive potential when these epitopes are similar to self-epitopes, and an immune response will be directed against host-antigens as well. (**E**) Autoreactive CD8 T cells that recognize viral antigens through MHC-I may directly cause tissue damage when these epitopes are similar to self-epitopes.

**Figure 3 biomedicines-11-01937-f003:**
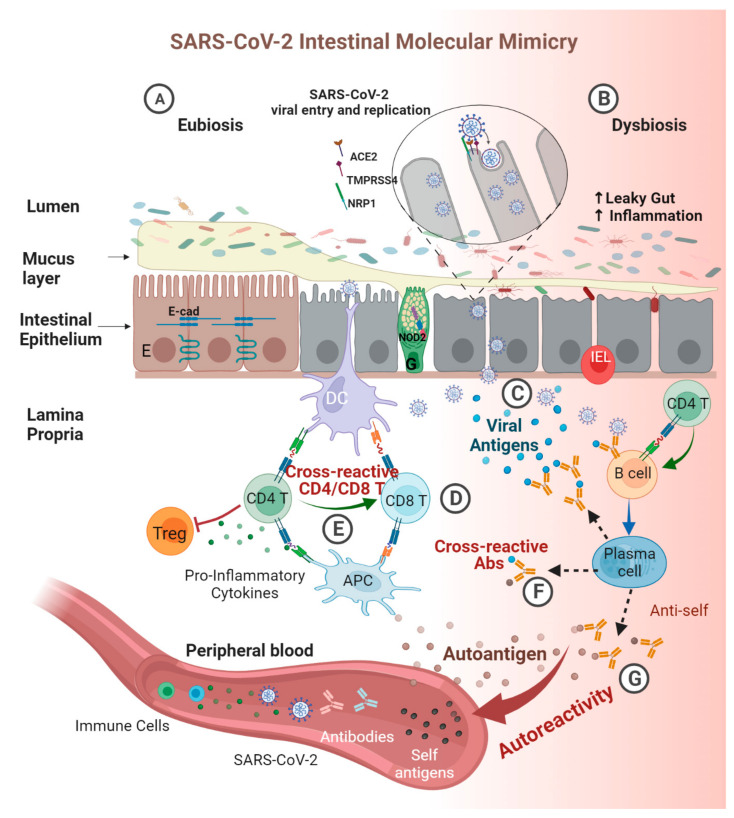
Schematic presentation of sequence similarity leading to cross-reactivity between SARS-CoV-2 and self-antigens implicated in gut-associated ADs. (**A**) During homeostasis, the epithelial cells interact together and maintain tight junction functional integrity. (**B**) Epithelial cells that express ACE2 are exposed to SARS-CoV-2 infection. Upon cellular penetration, intestinal homeostasis is compromised. The damaged epithelium enhances intestinal permeability and increases local and systemic inflammation, resulting in hyperstimulation of the immune responses. (**C**) Viral antigens are detected and processed by the immune cells, and antigen-presenting cells deliver them to lymphoid tissues. (**D**) When adaptive immune cells are activated, they proliferate and migrate to the site of infection, trying to eliminate the invasive pathogen. (**E**) Cross-reactivity at the T cell level involves the recognition of viral epitopes and similar self-epitopes by the same CD4/CD8 T cell. (**F**) Cross-reactivity at the B cell level occurs when clonal antibodies bind to viral epitopes that are similar to self-epitopes. (**G**) As the viral infection spreads into enteric organs, effector T and B cells extend their defense as well. Thus, increasing their likelihood to encounter self-antigens with similar epitopes and invoking an auto-reactive attack.

**Table 1 biomedicines-11-01937-t001:** Sequence similarity between SARS-CoV-2 proteins and enteric self-antigens related to ADs.

Human Protein (Autoimmune-Related Disease) ^1^	SARS-CoV-2 Protein	Human vs. SARS-CoV-2	Ln	Identity %	Similarity %	Score ^2^
**Cytochrome P450 2D6** **(AIH, CD)**	Membrane glycoprotein	KKSLEQW	7	85.7	85.7	33
		KKLLEQW				
**Keratin,** **type I cytoskeletal 18** **(AIH)**	Spike glycoprotein	VSETNDTK	8	75	75	26
		VSGTNGTK				
	Spike glycoprotein	EELDKYW	7	85.7	100	33
		EELDKYF				
	nucleocapsid phosphoprotein	NARIVLQI	8	75	87.5	28
		NAAIVLQL				
**Myosin-11** **(AIH)**	Envelope small membrane protein	VKNDNSSR	8	87.5	87.5	30
		VKNLNSSR				
**Actin, alpha skeletal muscle (AIH)**	Spike glycoprotein	GDGVTHNV	8	75	75	28
		GIGVTQNV				
**Pyruvate dehydrogenase E1 component subunit alpha, somatic form, mitochondrial (PBC)**	Spike glycoprotein	ATRFAAAY	8	75	87.5	32
		ATRFASVY				
**Dihydrolipoyllysine-residue acetyltransferase component of pyruvate dehydrogenase complex, mitochondrial (PBC)**	Spike glycoprotein	PATPAGPK	8	75	75	32
		PATVCGPK				
	Spike glycoprotein	DVPIGAIIC	9	77.8	88.9	39
		DIPIGAGIC				
**Histone H1.0 (UC)**	Nucleocapsid phosphoprotein	PKKAKKPK	8	75	75	29
		PKKDKKKK				

^1^ Human proteins that are implicated in enteric autoimmune diseases: autoimmune hepatitis (AIH), celiac disease (CD), primary biliary cholangitis (PBC), and ulcerative colitis (UC). ^2^ Score is based on BLOSUM62 substitution matrix between amino acids.

**Table 2 biomedicines-11-01937-t002:** The functionality of enteric antigens and their corresponding similar sequences.

Similar Peptides	Enteric Protein and Potential Function/Pathogenesis	AD ^1^	Ref.
**KKSLEQW**	**Cytochrome P450 2D6 (CYP2D6, UniProt:P10635)** Has been recognized as the major autoantigen in type 2 AIH. In patients with AIH-2, the target for anti-LKM-1 antibodies has been identified as the 2D6 isoform of the large cytochrome P450 enzyme family.	AIH CD	[62,63,64,65]
**VSETNDTK** **EELDKYW** **NARIVLQI**	**Keratin, type I cytoskeletal 18 (Keratin-18, UniProt:P05783)** Keratin 18 (K18) is an intermediate filament protein whose phosphorylation/transamidation associates with its aggregation in Mallory-Denk bodies found in patients with various liver diseases: cirrhosis, PBC, and AIH.	AIH PBC	[66,67]
**VKNDNSSR**	**Myosin-11 (SMMHC, UniProt:P35749)** Autoantibodies to non-muscle myosin heavy chain were reported in patients with chronic liver diseases. Patients presenting with ANA and/or smooth muscle antibodies (SMA) account for about 80% of cases of AIH.	AIH	[68,69]
**GDGVTHNV**	**Actin, alpha skeletal muscle (Alpha-actin-1, UniProt:P68133)** Anti-actin antibodies were the first anti-cytoskeleton autoantibodies described in liver diseases. They are considered a marker of AIH-1 and are also frequently detected in PBC.	AIH PBC	[70]
**ATRFAAAY**	**Pyruvate dehydrogenase E1 component subunit alpha, somatic form, mitochondrial (PDH-E1α, UniProt:P08559)** Autoantibodies against multiple antigens of the PDH complex are found in sera of more than 90% of patients with PBC, in particular against the PDH-E1α subunit. Anti-mitochondrial autoantibodies (AMA) appear to be directed to a functional site of PDC-E1α inasmuch as they are able to inhibit enzyme function.	PBC	[71,72]
**PATPAGPK** **DVPIGAIIC**	**Dihydrolipoyllysine-residue acetyltransferase component of pyruvate dehydrogenase complex, mitochondrial (PDC-E2, UniProt:P10515)** PBC patients have been characterized to have autoreactive T-cell and B-cell responses directed at self-PDC-E2. The diagnosis of PBC is readily reached by the detection of specific AMA directed against PDH-E2.	PBC	[73,74]
**PKKAKKPK**	**Histone H1.0 (UniProt:P07305)** Histone H1 bears a recurring COOH-terminal epitope recognized by monoclonal ulcerative colitis-associated pANCA marker antibodies.	IBD	[75]

^1^ Autoimmune diseases: autoimmune hepatitis (AIH), celiac disease (CD), primary biliary cholangitis (PBC), ulcerative colitis (UC), and inflammatory bowel disease (IBD—including UC and Crohn’s disease).

**Table 3 biomedicines-11-01937-t003:** Similar epitopes’ sequence with immunoreactive validation in experimental assays in IEDB.

Human Epitope	SARS-CoV-2 Epitope	Human vs. SARS-CoV-2	IEDB Human Assays’ References	IEDB SARS-CoV-2 Assays’ References
**STLRNLGLGKKSLEQWVTEE**	EEL**KKLLEQW**NLVIG	KKSLEQW	Tcell [76]	Tcell [77]; Bcell(IgA) [78]
		KKLLEQW		
**VVSETNDTK**	WFHAIH**VSGTNGTK**RFD	VSETNDTK	HLA-C*06:02 [79]; HLA-B*27:09 [80]	HLA-I/II [81]; Tcell [77]; Bcell(IgM) [82]
		VSGTNGTK		
**EELDKYWSQ**	DSFK**EELDKYF**KNHT	EELDKYW	HLA-I [83]	Tcell [84]; Bcell(IgG) [85]; HLA-DRA*01:01/DRB1*04:01 [81]; Bcell(IgG1) [86]
		EELDKYF		
**NARIVLQI**	PA**NNAAIVLQL**PQGT	NARIVLQI	HLA-B*51:01 [87]	Tcell [88]; Bcell(IgM) [89]
		NAAIVLQL		
**NAKTVKNDNSSRFG**	R**VKNLNSSR**	VKNDNSSR	HLA-II [90]	HLA-A*30:01; Tcell [91]; HLA-A*01:01 [92]; Bcell(IgM) [89]
		VKNLNSSR		
**SGDGVTHNVPI**	QMAYRFN**GIGVTQNV**	GDGVTHNV	HLA-II [93]	Tcell [94]; Bcell(IgM) [89]; HLA-II [95]
		GIGVTQNV		
**EATRFAAAY**	**ATRFASVY**A	ATRFAAAY	HLA-B*15:02 [96]; HLA-B*44:02 [97]	HLA-A*30:01; Tcell [91]; HLA-II [95]; Bcell(IgG) [98]
		ATRFASVY		
**VPPTPQPLAPTPSAPCPATPAGPK**	VLSFELLHA**PATVCGPK**	PATPAGPK	HLA-DQ [99]; Bcell(IgG) [100]	HLA-DRA*01:01/DRB1*04:01 [81]; Tcell [101]; Bcell(IgG) [102]
		PATVCGPK		
**GTRDVPIGAIICITVGKPEDIEAFK**	SYEC**DIPIGAGIC**ASYQ	DVPIGAIIC	Bcell(IgG) [100]	HLA-I/II [81]; Tcell [94]; Bcell(IgG) [103]
		DIPIGAGIC		
**AATPKKAKKPKT**	TE**PKKDKKKKA**DETQ	PKKAKKPK	HLA-II [93]; HLA-DR [104]; HLA-DRB1*11:03 [93]	Tcell [88]; Bcell(IgG) [85]; Bcell(IgM) [82]
		PKKDKKKK		

## Data Availability

The data and software that supports the findings of this study are openly available in: (1) The Immune Epitope Database (IEDB) at www.iedb.org (accessed on 30 December 2022), reference number [145,146]. (2) UniProt Knowledgebase www.uniprot.org (accessed on 30 December 2022), reference number [54]. (3) Pairwise Local Alignment tool, EMBOSS Matcher, at www.emboss.sourceforge.net (accessed on 30 December 2022), reference number [55,147], A python script can be found at https://raw.githubusercontent.com/ebi-wp/webservice-clients/master/python/emboss_matcher.py (accessed on 30 December 2022).

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
