# Peer review of "SARS-CoV-2 Gut-Targeted Epitopes: Sequence Similarity and Cross-Reactivity Join Together for Molecular Mimicry"

_biomedicines, 2023, doi:10.3390/biomedicines11071937_

Round 1

Reviewer 1 Report

The manuscript deals with an important point regarding the molecular mimicry between SARS-CoV2 and the intestinal epitopes , I found the manuscript interesting, but I have the following comments

1- Clearify the purpose of the study and justify the contribution to the field

2- One limitation of the study, there is no wet lab work

3- SARS-CoV-2 uses the angiotensin-converting enzyme 2 (ACE2) and TMPRSS2 receptors which are expressed in GIT and this explains COVID-19 manifestation in the GIT, you have to consider that in the discussion

4- The study has to discuss possible reactivity between COVID-19 vaccines and the identified Similar epitopes’ sequences

Minor editing of English language required

Author Response

Thanks for your valuable comments and suggestions

Reviewer 2 Report

The authors study the alignment of amino acid sequences of SARS-CoV-2 with those of the human intestine. They would like to highlight the importance of the intestinal route in the genesis of pathologies related to SARS-CoV-2. The manuscript would have potential if the authors had also used biological supports such as cell or viral cultures. In my opinion a work only on the interaction between viral and human epitopes is not strong. Authors should strengthen their manuscript with data on biological systems.

Also not clear why the authors talk about autoimmunity. If it is a central theme it should be discussed in matches from the title.

Finally, how does their manuscript fit in the context of the viral variants described in recent years?

Minor editing

Author Response

Thanks for your valuable comments and corrections

Reviewer 3 Report

 General comment

Several previous articles have suggested that molecular mimicry between SARS-CoV-2 and human proteins drives autoimmunity.  The study delves into that hypothesis by exploring a large number of SARS-CoV-2 epitopes that might cross-react with antigens in the human gastrointestinal tract. Sequence alignment was performed among those viral epitopes and self-antigens implicated in enteric autoimmune conditions to elucidate possible cross-reactivity. SARS-CoV-2 epitopes were drawn from the Immune Epitope Database (IEDB), while enteric antigens were obtained from the UniProt Knowledge base. Bioinformatic search is well done, and the figures contained in the paper are very illustrative  

Similarities were found in four viral proteins and seven enteric antigens, related to ulcerative colitis, primary biliary cholangitis, celiac disease, and autoimmune hepatitis. In conclusion, these findings of cross-reactivity and sequence similarity between the SARS-CoV-2 viral proteins and human tissue antigens support the above-mentioned suggestion. Limits of the study are also well established, as the article deals with the interaction of SARS-CoV-2 and the human gastrointestinal tract. Potential mRNAs-based vaccination immune consequences are excluded.

The following minor points would be addressed before definitive acceptance

Abstract: Delete the number of words (190 W) at the end.

Discussion: eliminate the word “Authors” at the beginning.

Line 115: “A comparatively weak reaction was detected against human collagen and histone”. Please, indicate which collagen and which histone. There are several types.

Line 157: “In the corresponding author’s recent publications”. It would be helpful to be more precise. Please, indicate the most illustrative references about this point. Are those 30–35?

Author Response

Thanks for your valuable comments and sugestions

Round 2

Reviewer 1 Report

I am pleased to accept the manuscript in its present form

Reviewer 2 Report

I agree with my comments from the previous review. The authors have not improved the manuscript. Unfortunately, in my opinion the manuscript is devoid of novelty. I suggest complete and immediate rejection.